# Pathological Features in Paediatric Patients with TK2 Deficiency

**DOI:** 10.3390/ijms231911002

**Published:** 2022-09-20

**Authors:** Cristina Jou, Andres Nascimento, Anna Codina, Julio Montoya, Ester López-Gallardo, Sonia Emperador, Eduardo Ruiz-Pesini, Raquel Montero, Daniel Natera-de Benito, Carlos I. Ortez, Jesus Marquez, Maria V. Zelaya, Alfonso Gutierrez-Mata, Carmen Badosa, Laura Carrera-García, Jesica Expósito-Escudero, Monica Roldán, Yolanda Camara, Ramon Marti, Isidre Ferrer, Cecilia Jimenez-Mallebrera, Rafael Artuch

**Affiliations:** 1Pathology, Biobank, Pediatric Neurology, Neuromuscular Unit and Clinical Biochemistry Departments, Hospital Sant Joan de Déu and Institut de Recerca Sant Joan de Déu, 08950 Barcelona, Spain; 2Biomedical Center for Research in Rare Diseases CIBERER-ISCIII, 28029 Madrid, Spain; 3Instituto de Investigación Sanitaria de Aragón (IISA), Universidad de Zaragoza, 50011 Zaragoza, Spain; 4Department of Pathology, Complejo Hospitalario de Navarra-IdiSNA (Navarra Institute for Health Research), 31008 Pamplona, Spain; 5Pediatric Neurology Department, Hospital Nacional Niños “Dr Carlos Sáenz Herrera”, San José 267-1005, Costa Rica; 6Unitat de Microscòpia Confocal i Imatge Cel·lular, Servei de Medicina Genètica i Molecular, Institut Pediàtric de Malaties Rares (IPER), Hospital Sant Joan de Déu, Esplugues de Llobregat, 08950 Barcelona, Spain; 7Research Group on Neuromuscular and Mitochondrial Disorders, Vall d’Hebron Institut de Recerca, Universitat Autònoma de Barcelona, 08193 Barcelona, Spain; 8Department of Pathology and Experimental Therapeutics, University of Barcelona, 08007 Barcelona, Spain; 9Biomedical Center for Research in Neurodegenerative Diseases (CIBERNED), Bellvitge Institute of Biomedical Research (IDI-BELL), Hospitalet de Llobregat, 08007 Barcelona, Spain; 10Department of Genetics, Microbiology and Statistics, University of Barcelona, 08007 Barcelona, Spain

**Keywords:** TK2 deficiency, paediatric patients, ultrastructural studies, muscle biopsy, mitochondrial myopathies, ragged red fibres

## Abstract

Thymidine kinase (TK2) deficiency causes mitochondrial DNA depletion syndrome. We aimed to report the clinical, biochemical, genetic, histopathological, and ultrastructural features of a cohort of paediatric patients with TK2 deficiency. Mitochondrial DNA was isolated from muscle biopsies to assess depletions and deletions. The *TK2* genes were sequenced using Sanger sequencing from genomic DNA. All muscle biopsies presented ragged red fibres (RRFs), and the prevalence was greater in younger ages, along with an increase in succinate dehydrogenase (SDH) activity and cytochrome c oxidase (COX)-negative fibres. An endomysial inflammatory infiltrate was observed in younger patients and was accompanied by an overexpression of major histocompatibility complex type I (MHC I). The immunofluorescence study for complex I and IV showed a greater number of fibres than those that were visualized by COX staining. In the ultrastructural analysis, we found three major types of mitochondrial alterations, consisting of concentrically arranged lamellar cristae, electrodense granules, and intramitochondrial vacuoles. The pathological features in the muscle showed substantial differences in the youngest patients when compared with those that had a later onset of the disease. Additional ultrastructural features are described in the muscle biopsy, such as sarcomeric de-structuration in the youngest patients with a more severe phenotype.

## 1. Introduction

The mitochondria contains the most intricate metabolic pathways within the cellular metabolism. This complexity is partially explained by the double genetic origin that controls oxidative phosphorylation, where proteins are codified by either nuclear or mitochondrial DNA, and by the fundamental pathways that occur in mitochondria [1,2]. All manners of clinical presentations, ages of disease onset, and inheritance types are possible in mitochondrial diseases, which are a group of rare genetic disorders that impair different mitochondrial biological functions, including ATP biosynthesis [2,3]. Thus, mitochondrial diseases can be caused by mutations in nuclear and mitochondrial genes, and defects in over 300 genes are reported to cause mitochondrial disease [3].

Mitochondrial thymidine kinase (TK2) is a nuclear DNA-encoded mitochondrial enzyme that catalyses the phosphorylation of thymidine in mitochondria. Its function is essential for thymidine and deoxycytidine triphosphate (dTTP and dCTP) synthesis in noncycling cells to maintain and control mitochondrial DNA (mtDNA) replication. Thus, mutations in the TK2 gene cause mtDNA depletion or multiple mtDNA deletion syndromes [1], which has a deep impact on mitochondrial energy metabolism. Regarding the natural history of the disease, a large series of patients have been previously reported [4,5,6,7,8,9], and the clinical spectrum has been extensively described, with three main phenotypes: (i) infantile-onset myopathy with severe mtDNA depletion, frequent neurological involvement, and a rapid progression to early mortality; (ii) childhood-onset myopathy, with mtDNA depletion and a moderate-to-severe progression of generalized weakness; and (iii) late-onset myopathy, with mild limb weakness at onset and a progression to respiratory insufficiency [6]. Although skeletal muscle seems to be the target organ in TK2 deficiency, it has been suggested that tissues other than muscles may be involved, such as the brain, eyes, heart, and liver [10]. Muscle biopsies have traditionally been considered the gold standard for the analysis of multiple mtDNA deletions and general depletion, including those caused by the TK2 deficiency [11]. In all cases, the main histopathological features in TK2 deficiency are ragged-red fibres (RRFs), described in variable numbers with cytochrome C oxidase (COX) defects [6]. RRFs are hyperreactive muscular fibres with succinate dehydrogenase (SDH) reactions indicating mitochondrial proliferation, and they are usually COX-deficient [11]. To the best of our knowledge, the presence of a dystrophic pattern, myonecrosis, increased endomysial connective tissue, or inflammatory features have scarcely been reported in TK2 deficiency cases [6,12]. Ultrastructural features in the cohorts of TK2-deficient patients have not been published.

With this background, we aimed to report the histopathological, immunohistochemical, and ultrastructural features of a cohort of paediatric patients, including the infantile- and childhood-onset forms of the TK2 deficiency.

## 2. Results

### 2.1. Clinical, Biochemical, and Genetic Data of the Patients

The main clinical, biochemical, and genetic data are presented in Appendix A. We report on eight affected non-consanguineous children (five males and three females) from eight families. Two cases corresponded to the infantile-onset form (below 1 year of age) and the other six corresponded to the childhood-onset form (between 1 and 12 years old). The first symptoms occurred before the age of 36 months in all cases. Based on the clinical characteristics, we divided the patients into two subgroups: those with an onset before 2 years of age who showed a very rapid progression of symptoms (n = 5), and a second group that debuted after the age of 2, of which the patients showed a moderate progression, with a loss of ambulation before adolescence (n = 3). In infants younger than 24 months of age, the main symptom was failure to thrive, with feeding problems and severe muscular hypotonia. In children over 24 months of age, the clinical picture was more subtle with progressive weakness and facial involvement. The patients did not have dysmorphic features or associated malformations. No patient showed intellectual disability. Two of the youngest two-year-old patients (one was not treated) died. The rest of the patients were treated with oral pyrimidine deoxyribonucleosides (thymidine and deoxycytidine), and their clinical outcome has been previously reported, with all of them being alive [6,13].

Regarding the biochemical and genetic features, while the blood lactate was normal in most cases, the level of creatine kinase was elevated in all cases but one, as was the level of GDF-15 [14]. Other biochemical analyses disclosed normal results. The measurement of mtDNA content in the muscle was abnormal in most cases (from 55 to 90% of mtDNA depletion). Six out of eight patients presented an mtDNA percentage of depletion equal to or higher than 70%. Multiple deletions were also found in the two patients presenting lower percentages of depletion (case 6 and 8). All patients presented pathogenic mutations in the TK2 gene: five compound heterozygotes and three homozygotes. The nomenclature of the TK2 variants follows the numeration of the GenBank accession number NM_004614.4 sequence, where one corresponds to the A nucleotide of the initial ATG codon. All patients had previously reported pathogenic mutations [6,13]. The TK2 activity was frankly reduced in almost all patients for which cultured fibroblasts were available for this determination. Only in one case (case 6) was the reduction of TK2 activity mild. The substrate-saturating conditions for the TK2 activity determination that we have used are more sensitive for detecting reductions in the maximum velocity (Vmax) of the enzyme than for detecting alterations in the affinity for the substrates [15,16]. In fact, the enzyme kinetics of the homozygous mutation for this particular patient (p.His121Asn) markedly increases the Km for ATP [13], which is in agreement with this explanation.

### 2.2. Pathological Characteristics of Patients

The main pathological and ultrastructural findings are stated in Table 1 and Figure 1, Figure 2, Figure 3 and Figure 4.

The muscle histological and histochemical analysis showed two main patterns related to the age of onset regarding symptoms: one pattern displayed more severe features corresponding to patients in whom first symptoms were observed before the age of 2 years, and the other with a mild pathological involvement corresponding to three cases older than 24 months. (Figure 1). All muscle biopsies revealed a myopathic pattern with variable muscle fibre size and occasionally internal small myonuclei, as well as the presence of a variable number of RRFs. The number of RRFs and COX-negative fibres was greater at younger ages.

In general, an increase in SDH activity was observed in all muscle biopsies, indicating mitochondrial proliferation (Figure 1(1c,2c)). The fibres displayed small sarcolemmal vacuoles, which were observed as areas devoid of staining in the haematoxylin-eosin slides (Figure 1(1a)). These vacuoles showed a high lipid content. No endomysial or perimysal connective tissue were seen. The fibres displayed regenerative changes with an overexpression of neonatal myosin (Figure 1(1e)) and alteration in fibre typing with a predominance of type I muscle fibres; these findings were more intense in the infantile-onset patients.

Occasional endomysial and perimysial mononucleated inflammatory cells were observed, mainly corresponding to macrophages with myonecrosis, and they were accompanied by isolated lymphocytes. In the infantile-onset cases, the macrophagic inflammatory infiltrate was distributed in the perimysium, endomysium, and in relation to the muscle fibres, while in the childhood-onset cases this infiltration was limited to necrotic muscle fibres (Figure 2).

These findings are accompanied by an overexpression of MHC I in the sarcolemma of practically all of the fibres with sarcoplasmic expression in regenerative fibres (Figure 1(1f)). The oldest patients presented fewer regenerative fibres (Figure 1(2e)), and MHC I overexpression was identified in the isolated fibres (Figure 1(2f)).

### 2.3. Immunofluorescence Studies for Mitochondrial Complexes

Immunofluorescence studies specific to the mitochondrial respiratory chain complexes showed a marked reduction in both complexes, CI and CIV (Figure 3(1a,d,2a,d)), supporting the findings that we observed during the COX staining (Figure 1(1d,2d)). The mitochondrial mass was evaluated by means of porin immunostaining, showing a lower expression in TK2 patients (Figure 3(1b,e,2b,e)) when compared with healthy controls (Figure 3(3b,e). In the cases that presented a lower number of RRFs (cases 7 and 8), the analysis of immunofluorescence for CIV complexes showed a greater number of fibres, with a decrease in staining when compared with the COX activity. These cases corresponded to the childhood-onset patients in our series (Figure 3(2d)).

### 2.4. Ultrastructural Examination

An ultrastructural study was performed for four of the cases (Table 2 and Figure 4).

The main feature in all cases was the presence of a population of abnormal mitochondria. In three out of four patients (two cases with onset before 24 months of age), several muscle fibres demonstrated a distinct subsarcolemmal accumulation of abnormal mitochondria. Frequently, the mitochondria were markedly enlarged or had an abnormal morphology, and they exhibited densely packed, concentrically arranged lamellar cristae. Occasionally, the mitochondrial matrix displaced spaces with electron-dense, osmophilic granules. We found three major types of mitochondrial alterations, mainly consisting of concentrically arranged lamellar cristae (Figure 4(1c)), electron-dense granules (Figure 4(3a)), and intramitochondrial vacuoles with no clear membranes (Figure 4(3c)) and composed of fine granular material. Furthermore, in patients, fibres showed an accumulation of lipid droplets (Figure 4(1a,2a)) in association with altered mitochondria, and these findings were more pronounced in the infantile-onset. An electron microscopy examination also showed that the youngest case (case 1), when compared with the oldest case (case 8), had a greater involvement of the sarcomeric muscle fibre with respect to regenerative changes, autophagic vacuoles, and the alteration of the intermyofibrillar pattern with the presence of cytoplasmic bodies (Figure 4(1a)). The oldest case (case 8) showed near-normal ultrastructural results, with minor alterations of the myofibrils, including their focal disorganization (Figure 4(2c)).

## 3. Discussion

The clinical, biochemical, and molecular features have been consistently reported in TK2-deficient patients. Here, we report on a cohort of infantile-onset TK2-deficient patients, focusing on the histopathological findings. Some of the clinical and molecular data in this cohort of patients have been previously reported [6,13,14]. In our series, all patients presented the first symptoms of the disease before 36 months of age, but showed different degrees of severity and progression according to the age groups (onset before or after 24 months of age). It is important to note that with the possibility of treating patients, the natural history of this disease has changed, especially in the childhood-onset form, as described in previous works [6,8,9,14]. Thus, the histopathological and ultrastructural data of the muscle fibres can lead us to a better response capacity, and the data highlights the importance of starting this therapy before the muscular alterations are more difficult to reverse.

Histopathological features have been described in some TK2 deficiency cohorts [6,8,17], but a deep histological analysis including an ultrastructural study focused on paediatric patients has scarcely been performed. In infantile-onset patients, we observed regenerative fibres with myonecrosis. Regenerative changes with an overexpression of neonatal myosin and myonecrosis are frequently observed in muscular dystrophies [18]. However, this is an unusual feature in mitochondrial diseases [19]. On the other hand, there was no observed increase in endomysial or perimysial connective tissue, nor was there any increase in fat infiltration observed in the infantile-onset form of our TK2 patients, unlike what was reported in the series of adult patients [8]. These observations are relevant as they can help us determine differential diagnoses for muscular dystrophies in the paediatric population. Another finding in these patients, which is not typical of mitochondrial myopathies, is an alteration in fibre typing with a predominance of type I fibres, which are more commonly seen in congenital myopathies [20].

The pathological analysis of the COX fibres are one of the cornerstones in the diagnosis and research of mitochondrial diseases [11]. In our series of patients, the mitochondrial respiratory chain complex immunostaining findings reinforced the observations of the COX histochemistry stains. Furthermore, according to our results, the immunostaining seems to be more sensitive than classical COX histochemistry methods. The immunohistochemical detection of OXPHOS complexes represents a valuable additional diagnostic tool for the evaluation of mitochondrial diseases [21,22]. This is especially relevant when there are few RRF, since OXPHOS complex immunostaining reveals more fibres with CI and CIV depletion than what is observed with the COX histochemistry.

Another interesting difference in our TK2-deficient cohort of patients compared with other pediatric mitochondrial myopathies is that the inflammatory infiltrates that were present were mainly constituted by macrophages with few T-lymphocytes that were associated with an overexpression of MHC I, especially in younger patients, where inflammatory infiltrates are not related only to myonecrosis. Inflammatory features have scarcely been reported in mitochondrial diseases; with these findings, we confirmed the previous inflammatory features detected by our group in TK2-deficient patients using a transcriptomic analysis [12].

The electron microscopy examination in mitochondrial disease shows mitochondrial abnormalities, including an increase in number and size, and altered individual morphology, such as irregular cristae consisting of short and discontinuous folding throughout a mitochondrion’s matrix; furthermore, the examination depicts mitochondrial swelling, inclusion bodies, and paracrystalline inclusions [23,24]. Hypotheses arise to explain such abnormalities, such as the alteration in the cristae organization system leading to ultrastructural alterations such as concentric cristae or compartmentalization, among others. However, other pathomechanisms can explain some ultrastructural features in mitochondrial diseases, such as the activation of proinflammatory genes or the presence of compensatory responses to mitochondrial stress [23].

The ultrastructural examination of skeletal muscle biopsies from our TK2-deficient patients revealed an increase in the mitochondrial number and size, as well as their abnormal morphology, with densely packed, concentrically arranged lamellar cristae or concentric “onion-shaped” cristae. Occasionally, the mitochondrial matrix displayed spaces which contained electron-dense inclusions. In contrast to other mitochondrial disease pathological features, no paracrystalline inclusions or cristae linearization were observed in our cohort of patients [23,24]. Intramitochondrial vacuoles containing fine granular material also occurred in our series of samples. The artefact of fixation can be excluded, as the biopsies have been processed at different times. The content of the vacuoles mimics the fine granular material of the cytoplasm, but the internalization of the cytoplasm into the mitochondria is not clear, as it is not limited by a clear mitochondrial membrane. Interestingly, these newly reported features were only observed in the infantile-onset cases. Sarcomeric de-structuration could explain the severe clinical involvement in the youngest patients. We did not observe mitochondrial fusion images, despite it being associated as a response to mitochondrial stress caused by various reasons, including mitochondrial DNA depletion [25].

Lipid accumulation is another feature frequently observed in muscle biopsies with impaired mitochondrial energy metabolism [11]. In our cohort of TK2 patients, an increase in the amount of lipids was observed, as well as a connection between lipid droplets and abnormal mitochondria, supporting the role of mitochondria in lipid metabolism.

## 4. Materials and Methods

### 4.1. Patients and Control Subjects

The clinical, biochemical, genetic, histopathological, and ultrastructural data from the eight infantile-onset and childhood-onset form TK2-deficient patients were recorded. Appendix A summarises the clinical, laboratory, and genetic data. Table 1 and Table 2 summarises the pathological and ultrastructural characteristics of the muscle biopsy. Some of our cases have been previously reported [6,9,12,13]. Results were compared with muscular biopsies from age-matched control subjects (age range: 9 months–16 years).

### 4.2. Pathological Studies

The skeletal muscle biopsy was obtained by an open surgical biopsy of the quadriceps in the diagnostic process of the disease. The age of onset of the disease and the age at which the biopsy was performed are detailed in Table 1. The skeletal muscle samples were processed and stained according to standard protocols with haematoxylin and eosin (H&E), modified Gomori trichrome, succinate dehydrogenase (SDH), cytochrome c oxidase (COX), oil red O, neonatal, slow and fast myosin, CD68, and major histocompatibility complex type I (MHC I), as published previously [26]. A double immunofluorescence study was also carried out with the porin antibody (clone VDAC-1, dilution of 1/50. Abcam. Cambridge, EE.UU.) as a surrogate of mitochondrial mass, with a complex I subunit NDUFB8 antibody (CI) (clone 20E9DH10C12, dilution of 1/50. Thermo Fisher Scientific. Waltham, MA, USA, EE.UU) and complex IV subunit IV antibody (CIV) (clone 1D6E1A8, dilution of 1/300. Thermo Fisher Scientific. Waltham, EE.UU) of the mitochondrial respiratory chain. The sections were incubated overnight at 4 °C. After incubation with the primary antibodies, the sections were washed with 1% PBS-Tween and then incubated with the secondary antibodies Alexa 448 and Alexa 594 (dilution of 1/500. Thermo Fisher Scientific, EE.UU) for 2 h at room temperature and were protected from light. After secondary incubation, the 1% PBS-Tween washings were performed, and slides were mounted with the Flouromount mounting medium. The images were captured using a TCS SP8 confocal laser scanning microscope (Leica Microsystems GmbH, Mannheim, Germany).

For the electronic microscopic analysis, the tissue was fixed in 2.5% glutaraldehyde, washed in cacodylate buffer, fixed in 1% osmium tetroxide, and dehydrated in graded alcohols. The tissue was embedded in resin and semi-thin and ultra-thin sections were performed. Four cases were available to be examined and were photographed using transmission electronic microscopy (JEOL Model 1100).

### 4.3. Other Laboratory Studies

Biochemical analysis: The biomarkers of mitochondrial dysfunction in plasma (amino acids, lactate, pyruvate, alanine, and growth differentiation factor-15 (GDF-15)) were analysed with ultraperformance liquid chromatography-tandem mass spectrometry, automated spectrometric procedures, and ELISA methods, as previously reported [27,28].

Genetic diagnosis: DNA from muscle biopsies was prepared by standard methods: mtDNA depletions were analysed through real-time quantitative polymerase chain reaction (PCR), as previously reported [29,30], and mtDNA deletions were studied by long-PCR. Genomic DNA was isolated from peripheral blood, and the coding exons and intronic boundaries of the TK2 gene were sequenced by Sanger sequencing.

TK2 activity determination: The TK2 activity was determined in primary cultured fibroblasts obtained from skin biopsies. Cells were grown in high-glucose Dulbecco’s modified eagle’s medium (DMEM: 4.5 g/L glucose), supplemented with 2 mM L-glutamine, 100 U/mL penicillin and streptomycin, and 10% dialyzed fetal bovine serum (FBS: Invitrogen). After reaching confluence, the FBS was reduced to 0.1% to induce quiescence. Two weeks later, we obtained cell pellets for the TK2 activity determination as described in Camara et al. and used [5′-3H]5-(2-bromovinyl)-2′-deoxyuridine (Moravek biochemicals) as a substrate [16].

## 5. Conclusions

We report the histopathological and ultrastructural features of muscular biopsies in a series of paediatric patients with TK2 deficiency. The pathological features in the muscle showed substantial differences in the youngest patients (onset before 24 months of age) when compared with those that had a later onset of the disease. Additional ultrastructural features are described in the muscle biopsies, such as sarcomeric de-structuration in the youngest patient with a more severe phenotype.

## Figures and Tables

**Figure 1 ijms-23-11002-f001:**
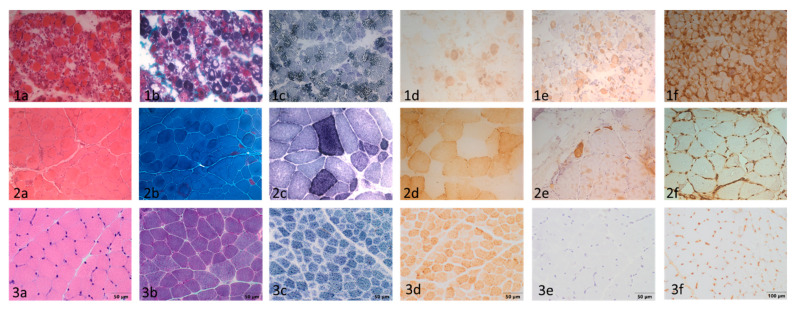
Muscular biopsy from patients with infantile-onset (row 1, corresponding to case 1), childhood-onset (row 2, corresponding to case 8), and a healthy control (row 3). Columns: (**a**) haematoxylin-eosin stain; (**b**) Gomori’s modified trichrome stain; (**c**) SDH stain; (**d**) COX stain; (**e**) immunohistochemical stain for neonatal myosin; and (**f**) immunohistochemical stain for MCH I. The infantile-onset patient showed a marked, unstructured muscular pattern with variability in the size of the fibres, internal myonuclei, and regenerative fibres. (**1a**) Many ragged red fibres and numerous fibres with large vacuoles; (**1b**) proliferation of mitochondria is visible as blue-coloured fibres; (**1c**) numerous fibres with COX depletion; (**1d**) small fibres with positive immunostaining for neonatal myosin indicated the presence of regenerative fibres; and (**1e**) MHC I is physiologically expressed in the capillaries. The patient presents a staining of the sarcoplasm and the sarcolemma in all the muscle fibres. Scale bar: 100 µm. The childhood-onset patient shows a more conserved pattern with a moderate variability in the measurement of fibres, with the presence of a population of hypotrophic fibres. (**2a**) Isolated ragged red fibres without subsarcolemmal reinforcements. No increased endomysial or perimysial connective tissue; (**2b**) occasional fibres with mitochondrial proliferation; (**2c**) isolated COX-negative fibres; (**2d**) very occasional regenerative fibres visible with immunohistochemical staining for neonatal myosin; (**2e**) mild MCH type I overexpression; and (**2f**) scale bar: 100–200 μm. (**3a**–**f**) Muscular biopsy from a healthy control. Scale bar: 50–100 μm.

**Figure 2 ijms-23-11002-f002:**
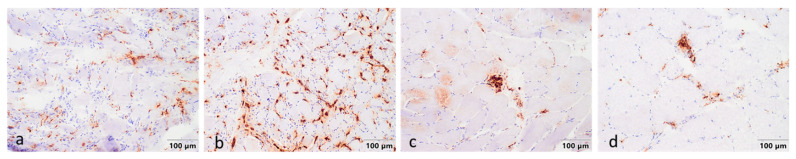
Immunohistochemical studies for CD68 in case 2 (**a**), 4 (**b**), 7 (**c**), and 8 (**d**). The infantile-onset patients (**a**,**b**) presented a diffused inflammatory infiltration in the endomysium and perimysium, while the childhood-onset cases (**c**,**d**) have CD68-positive inflammatory cells isolated in the endomysium mostly related to myonecrosis. Scale bar: 100 µm.

**Figure 3 ijms-23-11002-f003:**
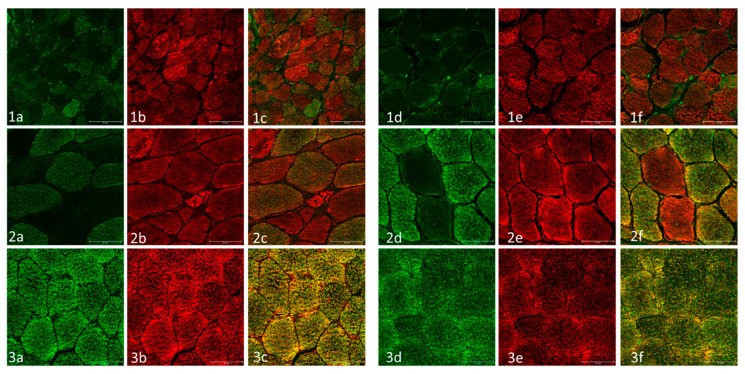
Muscular biopsy from patients with infantile-onset (row 1, corresponding to case 1), childhood-onset (row 2, corresponding to case 8), and a healthy control (row 3). Columns: (**a**) mitochondrial complex I immunofluorescence stain; (**b**) mitochondrial porin immunofluorescence stain (surrogate for mitochondrial mass); (**c**) merge (mitochondrial co-localization); (**d**) mitochondrial complex IV immunofluorescence stain; (**e**) mitochondrial porin immunofluorescence stain; and (**f**) mitochondrial co-localization. In the patient with infantile-onset, a severe deficiency of CI and CIV are shown in almost all the muscular fibres. Red fibres are present in the co-localization images, indicating a deficiency in mitochondrial complexes. In the childhood-onset case, a decrease in staining intensity are observed with the presence of some negative fibres, which are labelled in red in the co-localization image. In the healthy control, the mitochondrial complexes and porin have normal expression and the co-localization is coloured yellow, showing a correct co-expression of both proteins. Scale bar: 50 µm.

**Figure 4 ijms-23-11002-f004:**
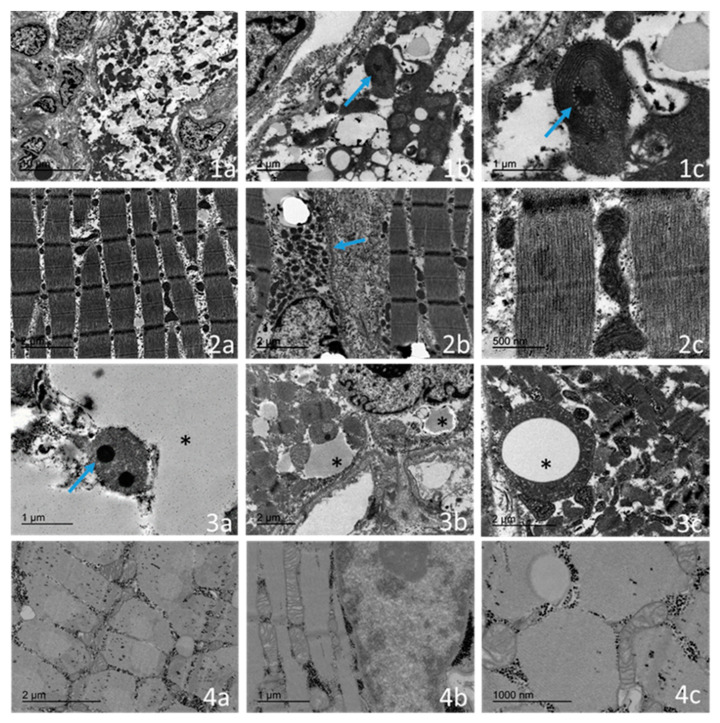
Transmission electron micrograph of the infantile-onset form (case 1) in the skeletal muscle region, showing a marked distortion of the intermyofibrillary pattern with an increase in the number of mitochondria and content of lipids (**1a**: scale bar of 10 µm was chosen to better observe a single muscular fibre). The mitochondria are enlarged with abnormal cristae (blue arrow) (**1b**: scale bar of 2 µm) and concentric, “onion-shaped” cristae with electrodense granule (blue arrow) (**1c**: scale bar of 1 µm was chosen to detect the electrodense inclusions that are not appreciable with lower magnification). The childhood-onset form (case 7) shows a relative conserved sarcomeric structure with a mild increased lipid content and lipid droplet size (**2a**: scale bar of 2 µm). Mitochondria are arranged around the nucleus with a slight increase in quantity (blue arrow) (**2b**: scale bar of 2 µm). The isolated mitochondria show mild abnormalities (**2c**: scale bar of 0.5 µm was chosen to rule out the presence of electrodense inclusions and intramitochondrial vacuoles). In the other infantile-onset patient (case 4), we observed intramitochondrial electrodense inclusions (blue arrow) (**3a**: scale bar of 1 µm); there is an intimate relationship between the mitochondria and lipid droplets (star) (**3b**: scale bar of 2 µm), and the intramitochondrial vacuoles contain fine granular material (star) (**3c**: scale bar on 2 µm). The healthy control child showed normal-sized mitochondria with unaltered cristae networks (**4a**: scale bar of 2 µm; **4b**: scale bar of 1 µm; **4c**: scale bar of 1 µm). Electrodense inclusions are not appreciable at a 1 µm scale bar.

**Table 1 ijms-23-11002-t001:** Histopathological (n = 8) features in the muscle biopsies from paediatric patients with TK2 deficiency.

Case	1	2	3	4	5	6	7	8
% RRF	70%	30%	40%	40%	25%	10%	8%	5%
% SDH	20%	12%	15%	5%	10%	10%	3%	8%
% COX negative	90%	65%	60%	90%	70%	25%	14%	5%
Increased lipid content	Yes	Yes	Yes	N.A.	Yes	Yes	Yes	No
Cytoplasmic bodies	No	No	Yes	Yes	Yes	No	No	No
Expression of neonatal myosin	20%	25%	20%	25%	N.A.	N.A.	4%	2%
Sarcolemmal vacuoles	Yes	Yes	No	Yes	Yes	Yes	Yes	No
Inflammation	Severe	Severe	Mild	Severe	Moderate	Mild	Mild	Mild
Macrophages infiltration	Yes	Yes	Yes	Yes	N.A.	N.A.	Yes	Yes
T-lymphocyte infiltration	Isolated	Isolated	Isolated	No	N.A.	N.A.	No	Isolated
B-lymphocyte infiltration	No	Isolated	No	Yes	N.A.	N.A.	Yes	No
MHC I overexpression	Yes	Yes	Yes	Yes	N.A.	N.A.	Yes	Yes
Myofagocytosis	Yes	Yes	Yes	Yes	No	No	Yes	Yes
Variation size fibres	Yes	Yes	Yes	Yes	Yes	Yes	Yes	Yes
Predominate type I fibres	Yes	No	Yes	Yes	N.A.	N.A.	Yes	Yes

**Table 2 ijms-23-11002-t002:** Ultrastructural features (n = 4) in the muscle biopsies from the paediatric patients with TK2 deficiency.

Case	1	4	7	8
Electron microscopy				
Increase in mitochondrial number	Yes	Yes	Yes	Yes
Changes in size and shape	Yes	Yes	Yes	Yes
Anomalous cristae	Yes	Yes	Yes	Yes
Presence of electron-dense inclusions	No	Yes	No	No
Lipid accumulation	Yes	Yes	Yes	No
Atrophic fibres	Yes	No	Yes	No
Cytoplasmatic bodies	Yes	Yes	No	No
Autophagic vacuoles	Yes	No	Yes	No

## Data Availability

Data are available upon reasonable request. Deidentified patient data.

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
