# Peer review of "Pathological Features in Paediatric Patients with TK2 Deficiency"

_ijms, 2022, doi:10.3390/ijms231911002_

Round 1

Reviewer 1 Report

In figure 1, Authors reported histopathological study by Gomori's modified trichrome stain, SDH stain, COX stain. These staining were previously reported in pediatric myopathy with TK2 mutation.

In figure 4, the electron microscopy of skeletal muscle showing mitochondrial abnormality, abnormal lipid accumulation. Visually understanding of different features from EM images is not clear. 

 In Materials and Methods (line 321-338), authors mentioned for Biochemical analysis, Genetic diagnosis, in vitro TK2 activity determination however, the result of these studies were not included in this manuscript.

More pathological features of mitochondrial abnormalities should be included, e.g., immunostaining of mitochondrial protein to determine the mitochondrial mass and number.

Author Response

Reviewer 1

In figure 1, Authors reported histopathological study by Gomori's modified trichrome stain, SDH stain, COX stain. These staining were previously reported in pediatric myopathy with TK2 mutation.

Yes, we agree with your comments, but the idea for adding this figure was to reinforce the other histopathological features and furthermore, such stains have scarcely been reported in early onset TK2 deficient patients.

In figure 4, the electron microscopy of skeletal muscle showing mitochondrial abnormality, abnormal lipid accumulation. Visually understanding of different features from EM images is not clear. 

Thank you for this comment. We agree with you and now we have marked with arrows and whiskers the main features from EM images and produced the image with higher magnification, as suggested by the other reviewer.

 In Materials and Methods (line 321-338), authors mentioned for Biochemical analysis, Genetic diagnosis, in vitro TK2 activity determination however, the result of these studies were not included in this manuscript.

We stated in table 1 the GDF-15 values, CPK, genetic data and TK2 activity results. We added in the result sections that the other biochemical analysis disclosed normal results.

More pathological features of mitochondrial abnormalities should be included, e.g., immunostaining of mitochondrial protein to determine the mitochondrial mass and number.

In figure 3 (column b), mitochondrial porin immunofluorescence stain is presented as a surrogate for mitochondrial mass. We have added a sentence in the result section to better explain these histopathological features. Line 170. “Mitochondrial mass evaluated by means of Porin immunostaining showed lower values in TK2 patients (1b,e and 2b,e) when compared with healthy controls (3b,e).”

Reviewer 2 Report

As a reviewer I would like to congratulate the authors for the project. Collectively, manuscript is written very well but introduction is not sufficient for the potential readers with no knowledge about the topic. Being said that I have following recommendations to improve the manuscript. 

1. According to the authors, studies were carried out in children between <1 to 12 years of age. As authors claimed all of them had their initial symptoms before 36 months (3 years). Did the authors observed children for 9 years (especially in 12 year old children)?

2. In page 3, lines 106-110 I suggest the authors to rewrite this sentence clearly. Also authors cited a paper from 2003,  can authors cite a more recent paper for the procedure. 

3. A serious attention is needed in alignment of Table 1. please fix the table for better readability. In-fact in current status this table is more confusing.

4. In Table 1, Along with stage of biopsy collection, addition of actual age of the children when biopsy samples collected will be more informative to the reader. 

5. Authors didn't mention the age of healthy controls used in the manuscript? 

6. In page 7-8, in electron microscopic studies, Is there a reason authors avoided the use of a healthy control in the electron microscopy studies?  

7. In the context of electron microscopy, in response to mitochondrial stress caused by various reasons (including mitochondrial DNA depletion), mitochondria tend to fuse together to share the DNA and this particular changes can be easily observed in electron microscopic images. Did authors observe mitofusion (mitochondrial fusion) in their images? 

8. In electron microscopic figure, please include the control images.

9. Fig 4 In print version, visual changes in mitochondrial matrix, etc are not clear. If possible, Authors are suggested to produce images with higher magnification.   

Author Response

Reviewer 2

As a reviewer I would like to congratulate the authors for the project. Collectively, manuscript is written very well but introduction is not sufficient for the potential readers with no knowledge about the topic. Being said that I have following recommendations to improve the manuscript. 

Thanks for your comments. We agree with you and we have extended the introduction of the manuscript regarding mitochondrial disorders.

According to the authors, studies were carried out in children between <1 to 12 years of age. As authors claimed all of them had their initial symptoms before 36 months (3 years). Did the authors observed children for 9 years (especially in 12 year old children)?

Thanks for this interesting comment. However, the oldest patients (case 7 and 8) were first visited in other Hospitals before 3 years of age, but final diagnosis and follow-up visits were done in our Hospital later on, between 8-12 years of age. So, data about evolution were not consistent and was not the aim of this work.

  1. In page 3, lines 106-110 I suggest the authors to rewrite this sentence clearly. Also authors cited a paper from 2003,  can authors cite a more recent paper for the procedure. 

We agree with you and we have modified the sentence. Together with reference 12, we have added reference 13, but we prefer to maintain also ref 12 since it is the original description of the mutational effect on Vmax values in TK2 enzyme.

  1. A serious attention is needed in alignment of Table 1. please fix the table for better readability. In-fact in current status this table is more confusing.

Thanks for this comment. In fact, in the previous version we had the table 1 was of better quality than the present one. We are submitting the table 1 in a separate file for editorial management.

  1. In Table 1, Along with stage of biopsy collection, addition of actual age of the children when biopsy samples collected will be more informative to the reader. 

We totally agree with you. We have added the age at biopsy for all patients in table 1.

  1. Authors didn't mention the age of healthy controls used in the manuscript? .

Thanks for your comment. We have added the age range of the healthy controls studied in the methods section (9 months-16 years).

  1. In page 7-8, in electron microscopic studies, Is there a reason authors avoided the use of a healthy control in the electron microscopy studies? 

In EM, the image interpretation usually does not need the analysis of a healthy control in parallel. The findings are compared with different EM image resources. However, we think that adding a new EM image from a healthy control can be very useful to facilitate interpretation. We have added it in figure 4 (see also point 8).   

  1. In the context of electron microscopy, in response to mitochondrial stress caused by various reasons (including mitochondrial DNA depletion), mitochondria tend to fuse together to share the DNA and this particular changes can be easily observed in electron microscopic images. Did authors observe mitofusion (mitochondrial fusion) in their images?.

No, we did not observe mitofusion. Since we think that your comment is worthy to be included, we have added a new sentence in the discussion section of the manuscript stating this interesting aspect (line 284) and a new reference.

  1. In electron microscopic figure, please include the control images.

We have added EM images from healthy controls (see age range of controls in methods section).

  1. Fig 4 In print version, visual changes in mitochondrial matrix, etc are not clear. If possible, Authors are suggested to produce images with higher magnification.

Thank you for this comment. We agree with you and now we have marked with arrows and whiskers the main features from EM images and produced the image with higher magnification.

Round 2

Reviewer 1 Report

General comments

The authors have responded to all comments and mentioned in the revised manuscript. However, I noticed some minor shortcoming for Figure 4 on page number 7. It is difficult to compare among case samples with different  scale bar (i.e., 1a, 2a, 3a and 4a). Reference to figure 4.3a-c, authors listed as "Others finding". I encourage the authors to mention the source of sample (case number).

Author Response

Thanks for your comment. Regarding the part of “other findings”, we have added the case identification (case 4) in the figure legend.

Regarding the different scale bars, we agree with you that comparison may be more difficult to do. However, the scale bar size and length depended of the fiber size, and we chose the best option depending on the findings we would like to highlight (the electronic microscope applies an automatic scale bar adjusting by the field size). For example, in case 1a, we have a 10 µm scale bar (after application of a 5,000 x magnification) to better observe one muscular fiber and to identify the sarcomeric destructuration. Case 2a and 4a had the same scale bar size (12,000 x magnification), while, in the case 3a we tried to demonstrate, with the highest magnification, the electrodense inclusions, that are not appreciable with a less magnification and are not present in controls as well. The other images were comparable, but the scale bar size was chose again depending on the fiber size and the findings we wanted to highlight. We have modified the Figure 4 legend including these new statements.  

Reviewer 2 Report

necessary corrections were introduced in the manuscript. Manuscript is acceptable in its current form. 

Author Response

Thank you very much